# Predicting the Risk for Pathological Fracture in Bone Metastases

**DOI:** 10.3390/curroncol32060309

**Published:** 2025-05-28

**Authors:** Pavlos Altsitzioglou, Shinji Tsukamoto, Costantino Errani, Yasuhito Tanaka, Andreas F. Mavrogenis

**Affiliations:** 1First Department of Orthopaedics, National and Kapodistrian University of Athens, School of Medicine, 15562 Athens, Greece; 2Department of Orthopaedic Surgery, Nara Medical University, Kashihara 634-8521, Japan; 3Department of Orthopaedic Oncology, IRCCS Istituto Ortopedico Rizzoli, 40136 Bologna, Italy

**Keywords:** bone metastases, pathological fractures, Mirels’ scoring system, CTRA

## Abstract

Approximately 50–70% of patients with advanced cancer will experience bone metastases. The link between metastatic lesions and pathological bones is especially troubling since more metastases mean a higher chance of painful fractures, which can reduce mobility and often require surgery. Therefore, fracture risk predictions are essential for managing patients with bone metastases. However, the current methods for evaluating the risk of fractures are subjective, with low predictive value. This paper highlights how there being no effective comprehensive models for grouping patients by fracture risk due to skeletal metastases makes it harder to create personalized treatment plans; presents the methods currently used for objective evaluations of the pathological fracture risk in bone metastases; and discusses their pros and cons.

## 1. Introduction

In oncology, the complex nature of metastatic bone disease causes problems because of the interactions of tumor cells with the bone environment leading to compounded issues. Among patients with advanced cancer, approximately 70% experience bone metastases, causing many complications that greatly impair their quality of life, including severe pain, hypercalcemia, and pathological fractures [1]. The association between metastatic disease and bone weakness is particularly troublesome, as the more metastases there are, the greater the likelihood of painful fractures, which often result in poor mobility and the need for surgery (Figure 1) [2]. Therefore, understanding the risk of fracture is essential in managing patients with bone metastases. This paper addresses the problem of there being no effective comprehensive model for separating patients based on the fracture risk due to bone metastases, making it difficult to develop individualized treatment plans.

In this study, our key aim was to discover and identify important clinical and radiological factors that influence the fracture risk in these patients with metastatic bone tumors and to explore how these factors interact in a predictive model built from an analysis of the past and future data. It is hoped that these findings can be applied to clinical practice, enabling oncologists to manage their patients using risk-evidence-based tools better [3]. Learning about these dynamics is also important for the academic purpose of furthering the discussion on personalized cancer care.

## 2. Methods

We searched PubMed for important works of literature reporting the results of studies which evaluated the risk of pathological fractures in patients with metastases in the long bones and summarized them. In other words, we performed a narrative review.

## 3. The Prediction of Pathological Fractures Using Imaging Findings Alone

The positive predictive value (PPV) and the negative predictive value (NPV) of bone cortex destruction >50% in predicting pathological fractures as a result of metastatic lesions in the upper extremities were 70% and 67%, respectively [4]. The same values for the femurs were 26–68% and 91–100%, respectively [4,5], while the PPV and NPV of axial cortical involvement greater than 30 mm in predicting pathological fractures as a result of metastatic lesions in the femurs were 19–23% and 95–97%, respectively [6,7,8].

## 4. The Prediction of Pathological Fractures Using Imaging Findings and Clinical Symptoms

Mirels’ scoring system, introduced in 1989, combined clinical and imaging criteria to assess fracture risk (Figure 2) [9]. The PPV and NPV of a Mirels’ score of 9 or more in predicting pathological fractures resulting from metastatic lesions in the upper extremities were 21% and 63%, respectively [10], while for metastatic lesions of the femurs, these scores were, respectively, 10–32% and 90–100% [5,6,11,12,13]. The Mirels’ score is often criticized because of its low positive predictive value and the possibility of unnecessary surgery [14]. This limitation has spurred research into more advanced methodologies that can capture the biomechanical properties of bone affected by metastatic lesions better.

## 5. Finite Element Analysis

Finite element analysis (FEA) using three-dimensional modeling of bone mechanics has shown promise for predicting fracture risk by integrating patient-specific data with biomechanical assessments (Figure 3) [8,11,13,15,16,17]. Detailed distributions of bone strain and stress can be generated using FE simulations and models, which allow for an essential understanding of the mechanical behavior of bone. The creation of these models is based on actual clinical images, and this approach is referred to as “patient-specific” modeling. The models are then used to identify parameters, primarily ultimate stresses and bone strength, with the ultimate goal of obtaining an assessment of the fracture risk [8,11,17]. The objective is to generate reliable numerical simulations that can reproduce the experimental biomechanics, first in the elastic phase and second in the plastic phase up to the fracture load. The geometry is obtained through image acquisition and is then transformed into a mesh (model). The geometry is discretized into smaller elements in order to solve the equations and thus to calculate the loads and displacements. The gray level of the image is converted into density [18] and from this into intrinsic material properties (primarily Young’s modulus) [19]. One-leg standing loading is considered to reflect the fracture risk related to metastatic disease.

Patient-specific FEA has demonstrated high accuracy in correlating predicted fracture loads with actual clinical events, increasing the value of incorporating detailed biomechanical assessments into clinical practice. The PPV and NPV of FEA in predicting the incidence of pathological fractures in femurs with metastatic lesions were 29–75% and 96–100%, respectively [8,11,13,15,16,17].

In spite of these advantages, the use of FEA entails a number of problems which are yet to be addressed. These include the specific properties of the material used in model validation, scrutiny of the mesh quality, the use of appropriate methods for the energy balance, and the reporting of all of these metrics [21,22,23]. In simulations and analyses, bone strength is estimated by applying various loading conditions to the model using specific boundary conditions and failure criteria. Therefore, it is critical to create valid numerical models [24,25]. In addition, only limited information is available regarding the mechanical and structural properties of a tumor and the surrounding bone [26]. In general, the material properties used in FEA are obtained from empirical studies that have investigated how the material behavior of healthy bone tissue is represented in computed tomography (CT) images. However, the composition and the resulting material behavior of bone metastatic tissue can greatly differ from that of healthy bone tissue, and consequently, the strength of the bone is likely to be altered [27]. Kaneko et al. demonstrated that cortical bone from patients with cancer in general may have reduced mechanical properties, including a lower modulus and compressive strength, compared to those of bone from non-cancerous donors [28]. Other authors believe that the remodeling process is different in metastatic bones (especially, for example, after radiotherapy) and consequently that their behavior in fatigue testing may also be different [29,30]. Other possible factors that may weaken bone, besides radiation therapy, include chemotherapy, sarcopenia, immunodeficiency, weight loss, and other factors which are often cancer-related. In addition, the bearing requirements of the affected bones depend on the body size of the patient and their weight, as well as their activity level [31,32] and the loading regimen [32,33]. Finally, the FEA must be performed by an operator with a certain level of technical expertise, and the image processing and analysis require reasonably sophisticated software; it has been estimated that generating and performing calculations for FEA takes about 8 h for just one metastatic lesion in a femur [34]. A certain level of automation would therefore be required to make this method effective in terms of its time and cost before it could function as a routine, clinically useful system for providing information to support decisions.

## 6. CT-Based Structural Rigidity Analyses

Stiffness is a structural property of bone. It is a measure of the resistance of the bone to deformation caused by bending, axial compression, or torsional loading and is a way to combine both the geometric and material properties of a bone into one variable [31,35]. Transaxial quantitative CT images taken continuously across the bone are used to noninvasively measure bending and axial and torsional stiffness and can then be used to determine progressive changes in the structural properties of the bone. Both the causative load and the location of a fracture can then be predicted from these results, based on the cross-section with minimal stiffness. By using composite beam theory combined with continuous transaxial CT images, it is possible to calculate the structural stiffness of the bone and, in turn, predict a threshold for fracture risk. Predicting fracture risk using this noninvasive 3D approach takes into account the material properties of the bone, as well as the shape and the location of an osteolytic lesion, and its biological activity and is known as CT-based structural rigidity analysis (CTRA) (Figure 3).

The PPV and NPV of CTRA in predicting the incidence of pathological fractures in metastatic long bone lesions have been reported to be 18–54% and 100%, respectively [12,36]. Accurate assessments of structural rigidity may improve a physician’s ability to identify potential fractures and monitor treatment progress, leading to more effective prevention and treatment strategies [34,37].

The advanced imaging techniques examined in this study provide a more comprehensive assessment by integrating biomechanical data with clinical assessments to represent the structural changes and mechanical stresses that occur in metastatic bone better [34,37]. In addition, these methodologies allow for a detailed analysis of how variables such as the tumor type, lesion location, and interaction with normal bone tissue contribute to fracture risk, which is often overlooked by traditional scoring systems [34,37]. Notably, CTRA’s superior ability to detect subtle changes in bone integrity advocates its incorporation into standard clinical assessments to reduce unnecessary surgery and improve overall patient outcomes [12,36]. Ultimately, these findings not only reinforce the advantages of advanced imaging techniques over the traditional methods but also lay the groundwork for future research focused on refining predictive models of the risk of fracture in patients with metastatic bone disease (Table 1).

Furthermore, a comparative analysis with FEA shows that while both methods are promising, CTRA stands out due to its simplicity and efficiency in the clinical setting; while FEA provides valuable three-dimensional modeling insights, it often requires more resources and may not be easily integrated into routine practice [37]. Therefore, the adoption of CTRA may streamline clinical workflows and enable timely and informed decision-making, ultimately reducing the incidence of preventable fractures and unnecessary surgeries [37] (Table 1).

The clinical implications of these findings are far-reaching. Enhanced fracture risk predictions not only facilitate more effective and personalized treatment strategies but also underscore the importance of a multidisciplinary approach in managing metastatic bone disease. By integrating advanced imaging techniques with standard oncologic evaluations, clinicians will be better equipped to tailor preventive and therapeutic interventions to addressing both the mechanical and biological aspects of metastatic lesions [38]. This integration is especially important when multiple lesions are present, as the cumulative impact on bone strength requires more subtle and aggressive management strategies [39].

The first problem with CTRA is that it does not reflect biological characteristics and possible changes over time. Therefore, it is unable to distinguish between necrotic bone and viable bone or between tumor tissue and non-tumor tissue. It is also impossible to predict whether treatment will result in bone healing or tumor progression. Therefore, patients who are thought to be at low risk of fracture should be followed using clinical and radiographic approaches and monitored for any changes [12]. Second, many clinicians are not yet familiar with the nuances of advanced imaging techniques, and standardized protocols for incorporating these tools into routine practice are still under development [40]. Third, the relatively small sample sizes and variability in clinical settings across studies suggest that further validation in larger multicenter trials is essential to fully establish the clinical utility and reliability of these advanced imaging modalities [41,42]. Future studies should also consider integrating these imaging techniques with emerging biomarker and molecular data to refine the predictive models further and enhance personalized patient care (Table 2).

## 7. Conclusions

The data presented here highlight the clear advantages of using FEA and CTRA in predicting fracture risk compared to the Mirels’ scoring system; CTRA is more sensitive and specific in assessing low-risk situations and identifies the fracture risk more effectively than traditional methods. This study also suggests the need for a change in the way fracture risk is assessed in metastatic disease. The incorporation of such advanced imaging techniques into routine clinical evaluations may improve the accuracy of fracture predictions and lead to better treatment planning, preventing serious complications in patients with metastatic lesions. It is imperative to build on these findings to promote more research, especially with regard to the use of CTRA in different types of cancer and treatment planning.

## 8. Future Directions

A machine learning composite model based on the CT attenuation values and radiomic features of the proximal femur has recently been reported to be able to accurately predict proximal femur fractures caused by osteoporosis [43]. Therefore, future research is needed to accurately predict pathological fractures caused by bone metastasis using machine learning and artificial intelligence.

N-terminal cross-linked telopeptide of type I collagen (NTX) is a peptide derived from the amino terminal end of mature, cross-linked, type I collagen, which is released during bone resorption. Elevated NTX levels in patients with bone lesions caused by multiple myeloma or solid tumors are associated with an increased incidence of skeletal-related events, including pathological fractures [44]. In addition to NTX levels, genomic analyses to identify factors correlated with the incidence of skeletal-related events are necessary.

Furthermore, enrolling a larger number of patients with diverse backgrounds to investigate the mechanisms underlying the differences in fracture risk may provide important insights into personalized treatment strategies. This study proposes the development of clinical guidelines integrating advanced imaging diagnostic systems into the standard treatment. By improving clinical decision-making, the use of innovative imaging diagnostic tools is proposed to address the existing gaps and contribute to improved management and outcomes for patients with metastatic bone disease. Ultimately, combining diagnostic accuracy, effective treatment methods, and a patient-centered approach will pave the way for future advancements in the management of pathological fractures associated with bone metastasis.

## Figures and Tables

**Figure 1 curroncol-32-00309-f001:**
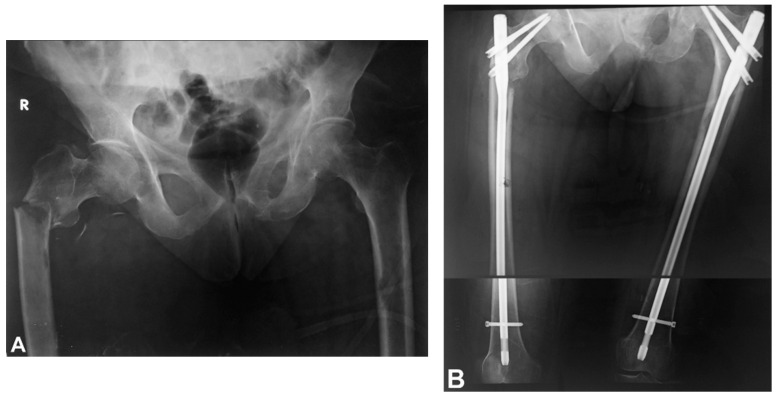
(**A**) An anteroposterior radiograph of the pelvis and hips of a woman aged 72 with metastatic breast cancer, showing a pathological subtrochanteric right femur fracture and metadiaphyseal osteolysis with an impending left femur fracture. She had been experiencing bilateral hip and thigh pain for the past 2 months. She was on chemotherapy and was given analgesics. (**B**) An anteroposterior radiograph of the femurs showing bilateral osteosynthesis with long, reconstruction-type intramedullary nails.

**Figure 2 curroncol-32-00309-f002:**
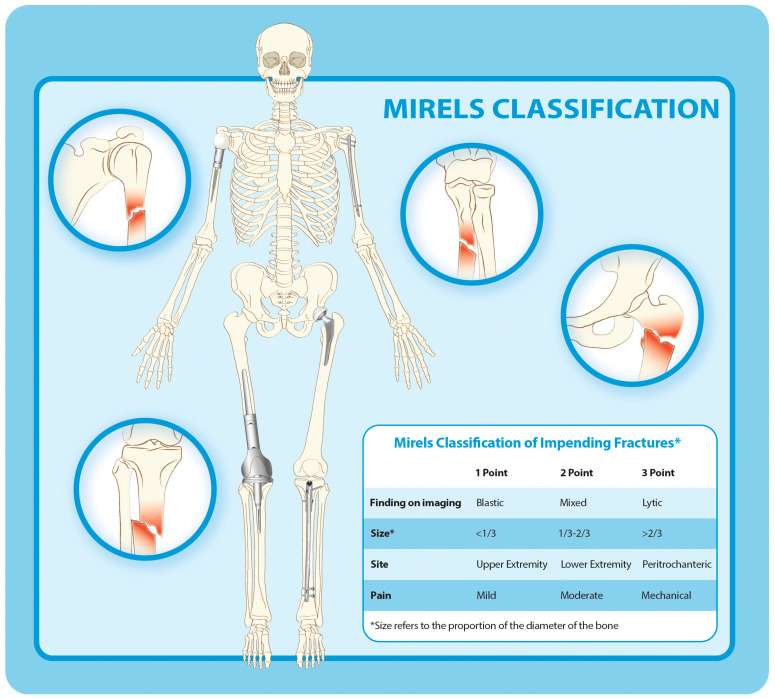
Mirels’ classification of impending pathological fractures.

**Figure 3 curroncol-32-00309-f003:**
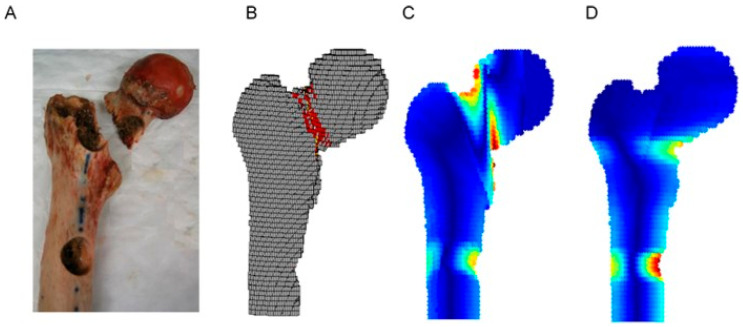
Fracture location as demonstrated by (**A**) mechanical testing, (**B**) a finite element analysis, and (**C**,**D**) computed tomography (CT)-based structural rigidity analyses (cited from Oftadeh et al.) [20].

**Table 1 curroncol-32-00309-t001:** Comparison of Mirels’ scoring systems, FEA, and CTRA for the prediction of pathological fracture risk.

Prediction Method	Imaging Method	Specialized Software	Analysis Time	Positive Predictive Value (%)	Negative Predictive Value (%)	Types of Loading	Anatomic/Modeling Limitations
Mirels’	Plane radiographs	No	<5 min	10–32	90–100	Not applicable	None
FEA	Computed tomography	Yes, to build the model and run an analysis	2–8 h, requiring engineering expertise	29–75	96–100	Functional loading (stance, gait, stair climbing, etc.)	Models and loading for the proximal femur different from those for the distal femur
CTRA	Computed tomography	Yes, to calculate section rigidities	<15 min, with custom software	18–54	100	Axial, bending, torsion	Errors associated with the ends of long bones

**Table 2 curroncol-32-00309-t002:** Key discussion points for the studies included in this review.

Discussion Topic	Key Insights
Importance of fracture risk prediction	Pathological fractures significantly impact quality of life, necessitating reliable prediction methods
Superiority of CTRA	CTRA shows greater positive predictive value compared to that of Mirels’ system, making it a more accurate tool for fracture risk assessments
Biomechanical integration	Unlike Mirels’ system, CTRA incorporates the mechanical properties of metastatic bone, improving the predictive accuracy
Clinical impact	Implementing CTRA in routine oncology practice may enhance early intervention, reducing unnecessary surgeries
Challenges & future directions	Further validation through multicenter trials is needed, along with integration of molecular biomarkers for personalized care

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
