# Peer review of "Predicting the Risk for Pathological Fracture in Bone Metastases"

_curroncol, 2025, doi:10.3390/curroncol32060309_

Round 1
Reviewer 1 Report
Comments and Suggestions for Authors
The authors review the clinical problems of metastatic bone tumors and predictive methods for their resolution such as Miles scoring system for their resolution, and they review previous literature regarding the usefulness of CTRA and FEA. The content is very interesting and important, and it's worth publishing.
The content is good enough as it is, but it would be even better if you could add the following points to deepen the reader's understanding.
・It would be easier for readers to understand if you provided a little more explanation, even if just briefly, about what CTRA and FEA are. If it is possible to explain it using a figure, please consider adding one. If the total number of figures and tables is increasing, you can delete the current Figure 1 as it is not very important.
Author Response
Responses to the Reviewer
Thank you for your detailed and thoughtful comments regarding our manuscript. We believe that our paper has markedly improved as a result of your valuable comments and feedback.
Reviewer 1
Comment
The authors review the clinical problems of metastatic bone tumors and predictive methods for their resolution such as Miles scoring system for their resolution, and they review previous literature regarding the usefulness of CTRA and FEA. The content is very interesting and important, and it's worth publishing.
The content is good enough as it is, but it would be even better if you could add the following points to deepen the reader's understanding.
・It would be easier for readers to understand if you provided a little more explanation, even if just briefly, about what CTRA and FEA are. If it is possible to explain it using a figure, please consider adding one. If the total number of figures and tables is increasing, you can delete the current Figure 1 as it is not very important.
Response
We created paragraphs of “5. Finite element analysis” and “6. CT-based Structural Rigidity Analysis” and added explanations for each. In addition, we added Figure 3, which explains FEA and CTRA.
Reviewer 2 Report
Comments and Suggestions for Authors
A review on the role of classification systems for predicting pathological fractures. The authors make the point that legacy classifications such as the one proposed by Mirel have low value, and newer techniques such as CTRA and FEA are clearly superior. They also elaborate on the advantages that CTRA has, which makes it the method of choice.
The manuscript is written in good English, but is verbose. The authors keep repeating the same meanings throughout the paper, which feels could be approximately one third in length without losing any information. The Figures chosen, especially Figure 2, are also redundant. The reader is more interested in the presentation of the 2 techniques (CTRA, FEA), with representative clinical cases showing their value. I believe the article should be rewritten across these lines.
Author Response
Responses to the Reviewer
Thank you for your detailed and thoughtful comments regarding our manuscript. We believe that our paper has markedly improved as a result of your valuable comments and feedback.
Reviewer 2
Comment
A review on the role of classification systems for predicting pathological fractures. The authors make the point that legacy classifications such as the one proposed by Mirel have low value, and newer techniques such as CTRA and FEA are clearly superior. They also elaborate on the advantages that CTRA has, which makes it the method of choice.
The manuscript is written in good English, but is verbose. The authors keep repeating the same meanings throughout the paper, which feels could be approximately one third in length without losing any information. The Figures chosen, especially Figure 2, are also redundant. The reader is more interested in the presentation of the 2 techniques (CTRA, FEA), with representative clinical cases showing their value. I believe the article should be rewritten across these lines.
Response
All duplicates have been removed throughout the manuscript. Figure 2 has been removed. We created paragraphs of “5. Finite element analysis” and “6. CT-based Structural Rigidity Analysis” and added explanations for each. In addition, we added Figure 3, which explains FEA and CTRA.
Reviewer 3 Report
Comments and Suggestions for Authors
This is a well-written and timely review that provides a comprehensive overview of current and emerging methodologies for assessing fracture risk in patients with bone metastases. The discussion of traditional methods such as Mirels scoring and their limitations, as well as the emphasis on advanced imaging techniques like CT-based structural rigidity analysis (CTRA) and finite element analysis (FEA), is well-founded and valuable to clinicians and researchers in the field. However, the manuscript would benefit from further refinement in several areas to enhance clarity, structure, and scientific rigor.
Here are the major comments :
- While the manuscript outlines the importance of predicting pathological fractures, it could better delineate between the goals of a narrative review versus a systematic review. Clarify in the Methodology section whether this was conducted as a narrative or systematic review. If systematic, include details about search strategy, inclusion/exclusion criteria, databases searched. And it would be necessary to provide the original data or table for all the studies being used in this review as supplementary materials.
- The results suggest CTRA has 100% sensitivity, which may be misleading without context. Please clarify the sample sizes and specific studies from which this figure is derived, and mention confidence intervals or variability to avoid overstatement.
- The section on future directions could be expanded to include recent efforts integrating machine learning, imaging-genomic correlation, or biomarkers with fracture risk prediction. This would give the review a forward-looking perspective that complements its current state-of-the-art analysis.
Minor issues:
- Some sections read more like a descriptive summary than academic prose. For instance, lines such as “This paper tackles the issue of there being no effective comprehensive model…” (lines 29–30) should be revised for clarity and formality.
- Repetition is noted in the discussion about the limitations of Mirels scoring and the superiority of CTRA across multiple sections. Consider consolidating these points to avoid redundancy.
- Ensure consistent use of terms such as “pathological fracture risk,” “impending fracture,” and “skeletal-related events.” Some sections may benefit from standardization of terminology to avoid confusion.
Author Response
Responses to the Reviewer
Thank you for your detailed and thoughtful comments regarding our manuscript. We believe that our paper has markedly improved as a result of your valuable comments and feedback.
Reviewer 3
Comment
This is a well-written and timely review that provides a comprehensive overview of current and emerging methodologies for assessing fracture risk in patients with bone metastases. The discussion of traditional methods such as Mirels scoring and their limitations, as well as the emphasis on advanced imaging techniques like CT-based structural rigidity analysis (CTRA) and finite element analysis (FEA), is well-founded and valuable to clinicians and researchers in the field. However, the manuscript would benefit from further refinement in several areas to enhance clarity, structure, and scientific rigor.
Here are the major comments :
While the manuscript outlines the importance of predicting pathological fractures, it could better delineate between the goals of a narrative review versus a systematic review. Clarify in the Methodology section whether this was conducted as a narrative or systematic review. If systematic, include details about search strategy, inclusion/exclusion criteria, databases searched. And it would be necessary to provide the original data or table for all the studies being used in this review as supplementary materials.
The results suggest CTRA has 100% sensitivity, which may be misleading without context. Please clarify the sample sizes and specific studies from which this figure is derived, and mention confidence intervals or variability to avoid overstatement.
The section on future directions could be expanded to include recent efforts integrating machine learning, imaging-genomic correlation, or biomarkers with fracture risk prediction. This would give the review a forward-looking perspective that complements its current state-of-the-art analysis.
Response
This paper is a narrative review. We replaced the previous sentence with “We searched PubMed for important literature reporting the results of studies which evaluated the risk of pathological fractures in patients with metastases to long bones, and summarized them. In other words, we performed a narrative review.” in the “Methods” section.
We replaced the previous sentences with “A machine learning composite model based on CT attenuation values and radiomic features of the proximal femur has recently been reported to accurately predict proximal femur fractures caused by osteoporosis [43]. Therefore, future research is needed to accurately predict pathological fractures caused by bone metastasis using machine learning and artificial intelligence.
N-terminal cross-linked telopeptide of type I collagen (NTX) is a peptide derived from the amino terminal end of mature, cross-linked, type I collagen, which is released during bone resorption. Elevated NTX levels in patients with bone lesions caused by multiple myeloma or solid tumors are associated with an increased incidence of skeletal-related events, including pathological fractures [44]. In addition to NTX levels, genomic analysis to identify factors correlated with the incidence of skeletal-related events is necessary.
Furthermore, enrolling a larger number of patients with diverse backgrounds to investigate the mechanisms underlying differences in fracture risk may provide important insights into personalized treatment strategies. This study proposes the development of clinical guidelines integrating advanced imaging diagnostic systems into standard treatment. By improving clinical decision-making, the use of innovative imaging diagnostic tools is proposed to address existing gaps and contribute to improved management and outcomes for patients with metastatic bone disease. Ultimately, combining diagnostic accuracy, effective treatment methods, and a patient-centered approach will pave the way for future advancements in the management of pathological fractures associated with bone metastasis.” in the “Future Direction” section.
Round 2
Reviewer 2 Report
Comments and Suggestions for Authors
I believe the revised version of the manuscript gives a clear view of the field and I find it suitable for publication.